



# A numerical modelling study of the physical mechanisms causing radiation to accelerate tropical cyclogenesis and cause diurnal cycles

Melville E. Nicholls[1], Warren P. Smith[1], Roger A. Pielke Sr.[1], Stephen M. Saleeby[2] and Norman B. Wood[3]

[1]Cooperative Institute for Research in Environmental Sciences, Department of Atmospheric and Oceanic Sciences, University of Colorado, Boulder, CO 80309, USA
[2]Department of Atmospheric Science, Colorado State University, Fort Collins, CO 80523, USA
[3]Space Science and Engineering Centre, University of Wisconsin, Madison, WI 53706, USA

*Correspondence to*: Melville E. Nicholls (Melville.Nicholls@colorado.edu)

**Abstract.** Numerical modeling studies indicate that radiative forcing can significantly accelerate tropical cyclogenesis. The primary mechanism appears to be nocturnal differential radiative forcing between a developing tropical disturbance and its relatively clear-sky surroundings. This generates weak ascent in the system core, which promotes enhanced convective activity. The goal of this study is to examine this hypothesis in more detail and in doing so shed light on the particular

physical mechanisms that are responsible for the accelerated development of the system. In order to clarify the effects of radiation the radiative forcing occurring in a full physics simulation is imposed as a forcing term on the thermodynamic equation in a simulation without microphysics, surface fluxes or radiation included. This gives insight into the radiatively induced circulations and the resultant changes to the temperature and moisture profiles in the system core that can influence convective development. Simulations with separate environment and core radiative forcings support the hypothesis that

differential radiative forcing due to nocturnal longwave cooling in the environment is the main factor responsible for accelerating the rate of tropical cyclogenesis. Simple idealized cloud experiments indicate that both cooling and moistening caused by the induced ascent significantly influence convective development, with the cooling having the largest impact. Diurnal cycles of Convective Available Potential Energy (CAPE), outgoing longwave radiation, deep convection and upper level outflows are examined and their relation to the radiative forcing is discussed.

## 1 Introduction

There have been considerable efforts to understand how longwave radiation and diurnally varying solar radiation affects oceanic tropical weather systems. One of the most influential early studies was by Gray and Jacobson (1977). Their observational analyses showed diurnal cycles were particularly evident in organized weather systems having deep convection. They found that at many places heavy rainfall was much greater in the morning than in the early evening.

Furthermore they hypothesized the reason for the diurnal cycle of deep convection results from day versus night variations in tropospheric radiative cooling between the weather system and its surrounding cloud-free environment. During the night a



significant difference between the radiative cooling rates in the surrounding cloud-free environmental air compared to that within the cloudy air of the weather system was thought to lead to increased low-level moisture convergence in the morning that favoured more intense convection and rainfall. However, early modelling studies did not lend much support to this hypothesis. Instead it was found that large-scale clear-sky cooling seemed to be the main factor aiding the convection by

continually destabilizing the troposphere (Dudhia, 1989; Miller and Frank, 1993; Fu et al. 1995). A study by Tao et al. (1996) also emphasized the role of large-scale clear-sky cooling, however their results indicated that Convective Available Potential Energy (CAPE) was not a significant factor and they proposed that the increase in relative humidity caused by the cooling was the main reason condensation occurred more readily in their simulations. Another mechanism that has been put forward to explain the diurnal cycles is changes to the thermal stratification aloft caused by cloud-top cooling and cloud base

warming (Cox and Griffith, 1979; Webster and Stephens, 1980; Xu and Randall, 1995).

Diurnal cycles of cirrus cloud cover have also been observed to occur in tropical cyclones (e.g. Browner et al., 1977; Muramatsu, 1983; Lajoie and Butterworth, 1984; Steranka et al., 1984; Kossin, 2002; Dunion et al. 2014). There has been some success in modelling tropical cyclone diurnal cycles (Navarro and Hakim, 2016; O'Neill et al. 2017; Dunion et al. 2019). However there still appears some uncertainty in the physical cause of diurnal cycles in these very large and intense

convective systems. A modelling study by Bu et al (2014) showed some interesting sensitivity of the structure of a simulated tropical cyclone to inclusion of cloud radiative forcing. Moreover cloud-radiative feedback may also impact tropical cyclone motion (Fovell et al., 2010).

Several cloud-resolving numerical modeling studies suggest that radiative forcing can significantly accelerate tropical cyclogenesis (Nicholls and Montgomery, 2013; Melhauser and Zhang, 2014; Nicholls, 2015; Tang and Zhang, 2016; Smith

et al., 2019). The study by Nicholls (2015; hereafter N15) emphasized that a major cause of this increased rate of development is nocturnal differential radiative forcing between the environment and the tropical disturbance that generates deep ascent in the core of the system. This is basically the same mechanism that was originally proposed by Gray and Jacobson (1977) to explain the diurnal cycles they observed in tropical oceanic cloud systems. While the radiatively induced ascent is very weak it can significantly enhance convective activity over the course of a night.

Numerical model experiments by N15 indicate that as an expansive cloud canopy forms aloft the longwave radiative cooling is reduced at low and middle levels. During the daytime there is not a very large differential radiative forcing between the environment and the tropical disturbance, but it becomes significant at night when in the absence of shortwave solar heating the longwave outgoing radiation results in a significant cooling of the environment. This induces a thermally driven transverse circulation characterized by subsidence in the environment and upward motion in the disturbance core. The

ascent results in a cooling tendency as well as upward advection of moist air leading to increased relative humidity throughout a deep layer in the system core at night. It was hypothesized that this is likely to aid in the triggering of convection, and provide a more favourable local environment at mid-levels for maintenance of buoyancy in convective cells due to a reduction of the detrimental effects of dry air entrainment. The cooling aloft would tend to increase CAPE which would favour convection. Additionally it was noted that there was a small increase in the surface tangential wind speeds





caused by the radiatively-driven circulation. N15 took a dynamical perspective of this mechanism by viewing it in terms of the lateral propagation of thermally driven gravity wave circulations (e.g. Nicholls et al., 1991; Mapes, 1993; Fovell 2002). Since the differential radiative forcing primarily induces a deep gravity wave mode that propagates rapidly at 40-45 m s$^{-1}$ the core of the tropical disturbance responds to changes in the radiative forcing of the environment quite quickly and throughout most of the depth of the troposphere. However it takes some time for the weak ascent to produce changes of the vertical profiles of vapour and temperature that are substantial enough to influence convective activity.

The study by N15 showed the existence of thermally-driven circulations for idealized specification of differential radiative forcing and also showed fields of the radiative forcing for a simulation of tropical cyclogenesis that suggested that they would induce similar circulations. The present study extends this previous work by first writing out the fields of radiative forcing to data files at a frequent time interval for a simulation of tropical cyclogenesis. These files are then read in during a new simulation that does not have the microphysics, surface flux and radiation schemes activated, and the radiative forcing fields are imposed as a source term in the model's thermodynamic equation. This allows the circulations induced by the radiative forcing in the full physics simulation to be isolated and their effects to be examined. In order to clarify how the changes in the vertical profiles of vapour and temperature are likely to influence convection several idealized cloud experiments are conducted that are initialized with a low-level warm and moist bubble. Additionally diurnal variations caused by the radiative forcing are examined. These include variations of Convective Available Potential Energy (CAPE), outgoing longwave radiation, deep convection and upper level outflows.

A difference between this study and N15 is that the system evolves into a tropical cyclone in a somewhat different manner. Nicholls and Montgomery (2013) found that even for a relatively simple initial condition consisting of a weak vortex with maximum winds at mid-levels over a warm sea surface and without any vertical wind shear that two distinct tropical cyclogenesis pathways occurred. One of the pathways termed pathway One showed an evolution similar to earlier studies conducted by Hendricks et al. (2004) and Montgomery et al. (2006). They emphasized the importance of vortical hot towers in the development of the tropical cyclone. A second pathway termed pathway Two showed similarities with the results of Nolan (2007). This pathway was characterized by a significant strengthening of the mid-level circulation and typically a contraction in size followed by the sudden formation of an intense small-scale vortex near the centre of the large-scale circulation. This involved an abrupt reduction in the radius of surface maximum winds. The N15 systems evolved along pathway Two and formed very small tropical cyclones, whereas in this study the tropical cyclone forms along pathway One and forms a larger tropical cyclone.

An outline of this manuscript is as follows: In section 2 the numerical model and the initial conditions for the simulation are described. The numerical experiments conducted are discussed in section 3. Results are presented in section 4 and conclusions are made in section 5.



## 2 Numerical model and initial conditions

RAMS is a nonhydrostatic numerical modeling system comprising time-dependent equations for velocity, nondimensional pressure perturbation, ice-liquid water potential temperature (Tripoli and Cotton, 1981), total water mixing ratio, and cloud microphysics, developed at Colorado State University (Pielke et al., 1992; Cotton et al., 2003). The microphysics scheme has

categories for cloud droplets, rain, pristine ice crystals, snow, aggregates, graupel and hail (Walko et al., 1995). For this study a two-moment version of the microphysics scheme s used, which predicts the number concentration of hydrometeors and enables the mean diameter to evolve (Meyers et al., 1997). Additionally a binned cloud-droplet riming scheme is employed (Saleeby and Cotton 2008). The radiation scheme solves the radiative transfer equations for the interaction of three solar and five infrared bands with three gaseous constituents, $H_2O$, $O_3$ and $CO_2$, and includes effects of the cloud

hydrometeors size spectra (Harrington, 1977; Harrington et al., 1999). The model has the capability of using multiple nested grids, which allows explicit representation of cloud-scale features within the finest grid while enabling a large domain size to be used, thereby minimizing the impact of lateral boundaries. Further details of the model physics used for this tropical cyclogenesis simulation can be found in N15.

The full physics simulation uses three grids with horizontal grid increments of 24, 6 and 2 km, and (x, y, z) dimensions of

170×170×58, 202×202×58, 353×353×58, respectively. Each finer scale grid is centred within the next coarsest grid. The vertical grid increment is 60 m at the surface and gradually stretched with height to the top of the domain at z = 24.3 km. A Rayleigh friction layer is included above z=18 km to damp upward propagating gravity waves.

Similarly to Montgomery et al (2006) the temperature structure is the mean Atlantic hurricane season sounding of Jordan (1958). The initial moisture profile is the same as used for the reduced CAPE experiments in that study, referred to as B2 and

B3. This profile has low level moisture reduced from the Jordan sounding by a maximum of 2 g kg$^{-1}$ at the surface (this is discussed in Nicholls and Montgomery, 2013). The vortex initialization is described in Montgomery et al. (2006). It is 8 km deep with maximum winds of 8 m s$^{-1}$ at a height of 4km and 4 m s$^{-1}$ at the surface. The radius of maximum winds is 125 km. The sea surface temperature is 28 °C. With this model configuration it takes three days for tropical cyclogenesis to occur and the system evolves along pathway One, as described in Nicholls and Montgomery (2013). Note that we have found that for

this current model configuration initial vortices that are large are more likely to evolve along this pathway than small initial vortices.

## 3 Description of experiments

Table 1 lists the thirteen experiments conducted in this study. The first is a full physics simulation of tropical cyclogenesis with radiation included. The second is otherwise identical to the first except the radiation scheme is turned off. For the first

experiment the radiative flux convergence was written out to data files at hourly intervals. For the third experiment data files containing these radiative flux convergence values at hourly intervals from the full physics simulation were read into a new simulation that had the radiation, microphysics and surface flux schemes turned off, and imposed as a source term on the


model's thermodynamic equation. Since the read in values of the radiative flux convergence were at hourly intervals a linear interpolation between hourly values was used to prevent step-wise jumps in the thermodynamic forcing. This simulation did not include a weak initial vortex. As shown by N15 the presence of a strong vortex can significantly impact the response to radiative forcing, however a test with the weak initial vortex used to initialize the full physics simulation showed relatively

minor differences. Since it is easier to interpret the perturbed fields from the basic state in the absence of the initial vortex we decided not to include it in the imposed radiative forcing experiments. Also to save computational expense these simulations did not include the third grid that has a 2 km horizontal grid increment.

    The objective of the fourth and fifth experiments is to ascertain how important the radiative forcing in the cloudy air of the tropical disturbance is compared to the radiative forcing in the relatively cloud-free environment. An estimate of the radius

of the cloudy air of the tropical disturbance was made as a function of time. For Experiment 4 the radiative forcing at distances less than this radius was set to zero, so this is referred to as the environmental forcing only case. For Experiment 5 the radiative forcing beyond this radius is set to zero, so is referred to as the system core forcing only case.

    Experiments 6-13 were conducted in order to obtain a better understanding of how convective cells in the full physics simulation were impacted by the changes to the vapour and temperature profiles in the core of the cloud system brought

about by radiative forcing. A low-level warm and moist bubble was used to initiate a cloud within a model domain having the same resolution as the fine grid full physics experiment, with the radiation and surface flux schemes deactivated, and without any initial winds. The initial sounding for Experiment 6 was obtained from the full physics simulation by horizontally averaging the temperature and vapour within a radius of 200 km from the centre of the domain at 57 h simulation time, which was at nightfall and approximately twelve hours prior to tropical cyclogenesis. The initial sounding

for Experiment 7 was obtained from the results for Experiment 3 with imposed radiation. A horizontally averaged sounding was obtained at 57 h and subtracted from a horizontally averaged sounding at 69 h. These changes in temperature and vapour due to the nocturnal radiative forcing were then added to the full physics averaged sounding at 57 h to produce a modified sounding that represented the changes that occurred due to radiative forcing throughout the night. Experiment 8 examined the effect of including the vapour change only, while Experiment 9 examined the effect of including the temperature change

only. For Experiment 10 the modifications were obtained from Experiment 4 that had turned off the environmental radiative forcing. For Experiment 11 the modifications were obtained from Experiment 5 that only included the environmental radiative forcing. Experiments 12 and 13 examined the effect of the radiative forcing during the previous daytime.

## 4 Results

### 4.1 General description of Experiment 1 and comparison with Experiment 2

Figure 1 shows time series of the minimum surface pressure, the total hydrometeor mixing ratio in the fine grid domain, the maximum azimuthally averaged tangential velocity at $z$=29.5 m, and the clear-sky surface shortwave radiation for the first experiment, and also includes comparison with the second experiment that does not have radiation. For Experiment 1 there is



a significant increase of the minimum surface pressure during the first 66 h, after which the pressure decreases rapidly to 983 hPa at 96 h. For the no radiation case there is only a small decrease in pressure by 96 h. It might seem unusual that for the simulation with radiation there is an increase in pressure initially. It occurs across the whole of the domain and is due to net cooling by longwave radiation. The standard version of RAMS omits terms in the prognostic pressure equation that allow

generation of compression waves by diabatic heating, and this can lead to mass trending if there is net diabatic heating or cooling in the domain as discussed by Nicholls and Pielke (2018). One way to prevent mass trending would be to include these terms and use lateral boundaries that were either walls or cyclic, so that mass could not leave the domain. For the radiation case the maximum tangential velocities increase rapidly at the end of the third day, increasing to 37 m s$^{-1}$ by 96 h, whereas for the no radiation case the maximum winds only reach 5 m s$^{-1}$ at this time. Even though the no radiation case was

run out to six days tropical cyclogenesis still did not occur.

The total hydrometeor mass for the whole of the domain shows a clear diurnal cycle for Experiment 1 with significant peaks occurring during the morning at 48 h and 70 h. Also shown in Fig. 1b is the hydrometeor mass within a radius of 200 km of the centre of the fine grid domain, and interestingly this shows a weak double peak between 60-84 h instead of a single large peak.

Figure 2 shows the y-component of velocity at 12 h intervals from 36-72 h through the centre of the domain for grid 2. At 36 h the low-level winds have begun to increase and by 48 h reach 9-10 m s$^{-1}$. At 60 h a prominent mid-level vortex is evident. Nicholls et al. (2018) demonstrated that sublimation of ice is a major factor in causing a mid-level vortex to form in these RAMS tropical cyclogenesis simulations. By 72 h winds at the surface have increased to over 12 m s$^{-1}$ and tropical cyclogenesis can be considered to have occurred. This evolution is similar to pathway One as described by Nicholls and

Montgomery (2013). We have noticed for these recent simulations, which have some slightly different physical parameterizations than this earlier study, that there is some sensitivity to the size of the initial vortex, with larger vortices being more likely to evolve along pathway One.

Figure 3 shows results at 45 h, which is at the end of the second night and just before the peak in total hydrometeor mass at 48 h seen in Fig. 1b. Fig. 3a shows a large radiative cooling occurring at the top of the canopy while at its sloping base

there is significant heating. In the surrounding environment cooling is occurring which is particularly strong around 8 km. The horizontal section of radiative flux convergence at z=6.5 km gives another perspective. There is less cooling to the east because the model includes the longitudinal variation of the shortwave radiation and the sun is beginning to rise. There is a hole in the cloud canopy that can be seen in Fig. 3d, which clearly influences the radiative flux convergence. Fig. 3c shows a strong outflow aloft and a weak inflow at low levels. Also a mid-level inflow is starting to develop. There has been

significant environmental cooling, which is particularly strong at z=8 km, as well as strong cooling at the top of the canopy (Fig. 3e). There is an annulus of high relative humidity in the troposphere primarily due to vertical moisture transport by convective towers, which tend to be concentrated in the region of the highest surface winds.

Results during the middle of the day at 52 h are shown in Figure 4. At this time the shortwave radiation in the surrounding environment of the tropical disturbance is tending to counteract the longwave cooling (Fig. 4a). Strong shortwave heating is





occurring near the top of the cloud canopy although at the very top the longwave cooling dominates. Within the bulk of the stratiform ice region there is a notable increase in radiative heating due to shortwave radiation compared with several hours earlier during the nighttime (Fig. 3a). Fig. 4c shows that the mid-level inflow has increased and this results in convergence of angular momentum and the development of the strong mid-level circulation seen a few hours later in Fig. 2c. Comparing Fig.

4d with Fig. 3d it is evident that convective cells have become less intense. The shortwave down in the environment decreases with height as would be expected since absorption is occurring (Fig. 4e). Within the centre there is still quite a significant gap in cloud cover, which allows more of the shortwave radiation to reach the surface than within the surrounding denser cloud area. Fig. 4f shows there is considerable reflection of shortwave radiation at the cloud top.

Figure 5 shows results 17 h later at 69 h, near the end of the following night. The horizontal extent of the cloud system has

grown significantly from 24 h earlier (Fig. 3). Again there is a significant horizontal gradient of the radiative forcing due to the longwave cooling in the environment. There are now far more convective cells. The potential temperature perturbations show very strong cooling at cloud top and also strong cooling in the environment at a height of 8 km. High humidities are apparent at mid-levels in the centre of the system. The horizontal cross-section of vertical velocity at a height of 3.6 km shows there are numerous convective-scale updrafts and downdrafts. The longwave down radiation field indicates that

within the cloudy air there is considerable emission towards the surface, while the longwave up radiation field indicates that the emission to space is considerably reduced above the cloud canopy.

**4.2 Imposed radiative forcing experiments**

The changes occurring in the first 36 h of Experiment 3 were to a large degree horizontally homogeneous. Significant cooling occurred in the troposphere from the surface to a height of 13 km. Temperature perturbations at low-levels were

approximately -1.5 K whereas between $z$=7 km to 9 km where longwave cooling was strongest they were approximately -3 K. These relatively cloud-free radatively induced temperature perturbations caused CAPE to increase from an initial value of 193 J kg$^{-1}$ to 426 J kg$^{-1}$ at 36 h. The cooling also caused significant changes to the relative humidity of approximately 5 % at low levels to over 10 % aloft where the cooling was strongest.

Fig. 6 shows azimuthally averaged fields about the centre of grid 2 for Experiment 3 at 40 h, which is in the middle of the

second night (Fig. 1d). Fig. 6a indicates a radial differential radiative forcing exists between the environment and the disturbance at low and mid levels, and this drives deep ascent at a radius of around 130 km (Fig. 6b). While only ~5 mm s$^{-1}$, it is still enough to produce significant deep moistening during the course of the night. At this time, the deep moisture anomaly seen in Fig. 6f reflects an increase in vapour of approximately 0.2 g kg$^{-1}$ compared with 6 h earlier (not shown). Fig. 6g shows an increase of relative humidity in the region of ascent. There has also been significant cooling at a height of

~8 km where it is approximately half a degree colder than 6 h earlier. These mesoscale changes to the vertical profiles of vapour and moisture brought about by radiative forcing are likely to be the main reason for the increase of deep convection during the nighttime seen in Fig. 1b. Fig. 6c shows there is an outflow at a height of 11 km and an inflow just above. The outflow looks like it is driven by the sloping region of infrared warming at the base of the cloud layer aloft causing a



prominent sloping region of ascent. There is also indication of an anticyclonic circulation developing in response to this outflow (Fig. 6d). The pressure perturbation field shows the surface pressure has increased significantly in response to net radiative cooling aloft and the pressure changes are approximately horizontally homogeneous with only small radial gradients discernible.

5       Figure 7 shows results at 52 h, which is in the middle of the following day. It is apparent that there is no longer any significant ascent at low levels although there is considerable ascent aloft. A strong outflow is still occurring aloft, but it is at larger radii and there is an inflow beneath it. There is now a significant anticyclonic circulation aloft associated with the radiatively driven outflow. Compared to 12 h earlier the temperature at a height of 8 km has cooled by another degree. Even during the daytime the temperature at this level cooled (not shown). It is quite horizontally homogeneous, so it is apparent

that the radiative heating during the daytime evident in Fig. 7a within the system core has not resulted in an increase in the static stability. However, comparing Fig. 6e and 7e it can be seen that higher up that the temperature at the top of the canopy has increased significantly presumably due to the large change in the radiative forcing at this height.

        N15 conducted an experiment where a heating function was applied that had prescribed cooling between the surface and 5 km and heating between 5 km and 10 km for a radius less than 200 km (Experiment 8). It was found that while the

temperature response initially showed weak low level cooling with weak warming above, that this situation did not last and temperature perturbations ended up being negligible after 12 h (Figure 10a of N15). This has some similarity to what is occurring in this case where the heating between 5 km and 10 km near the centre does not lead to appreciable temperature change since the heated air ascends and tends to rise along a dry adiabat, thereby cooling until it reaches the temperature of the surrounding air. Therefore care has to be taken not to assume that the diabatic heating in the core necessarily leads to a

significant temperature change that affects static stability. In addition to this response the radiative forcing shows that there is still some environmental cooling at upper levels, which can expected to contribute to forcing ascent in the system core and adiabatic cooling.

        The vapour perturbation field shows there has been a significant increase between 5 km and 10 km compared to 12 h earlier and while some of this occurred during the later part of the night it continued during the daytime, which is consistent

with the general ascent in this layer seen in Fig. 7b. This leads to a maximum in the relative humidity seen in Fig. 7g. Also there is a peak of water vapour near the surface in between air that is significantly drier. These low-level changes seem to be brought about earlier on by radiative cooling off the top of a shallow cloud layer, as well as low-level cooling in the centre where there is a gap in cloud cover, as can be seen in Fig. 6a and 3a. This appears to cause some weak subsidence and convergence and ascent in between which creates drying and moistening respectively.

While the low-level pressure perturbation has increased significantly compared to 12 h earlier, most of this increase occurred earlier on during the previous night. The daytime solar heating, which counteracts the longwave cooling, has significantly slowed down the rate of low-level pressure increase.

        Figure 8 shows results at 64 h, which is in the middle of the following night. Compared to 24 h earlier shown in Fig. 6, the system has grown extensively, but still shows a similar pattern of differential radiative forcing, which drives weak ascent in





the core of the system. There is now a very notable inflow at the top of the cloud canopy, which is causing a weak cyclonic flow. At the base of the cloud canopy the persistent outflow has resulted in a strong anticyclonic circulation at a radius of 600 km. Comparing with the anticyclonic flow aloft in Fig. 2c and d, it can be inferred that about half of the magnitude occurring in the full physics simulation may be a direct result of radiative forcing. Near the surface there is a cyclonic

circulation of ~2 m s$^{-1}$ at a radius of 140 km. The potential temperature perturbation shows significant cooling at the canopy top and at around a height of 8 km. The magnitude of the vapour perturbation now exceeds 1 g kg$^{-1}$ between middle and upper levels within a broad region of the system core. The relative humidity is also high in this region and at the canopy top where strong nighttime cooling has occurred. The low-level pressure has increased significantly compared to 6 h earlier.

        From the results of imposing the full physics radiative forcing in a simulation without microphysics and surface fluxes it

can be concluded that the accumulated thermodynamic and kinematic changes are quite large by 64 h. An interesting question is how much the changes are a result of radiative forcing in the relatively cloud-free environment versus in the cloud disturbance? To address this question Experiments 4 and 5 were conducted. A cloud canopy began to develop around 36 h, so up until that time both experiments used the same radiative forcing as for Experiment 3. The area of the cloud top increased approximately linearly between 36 h to 72 h from a radius of ~200 km to ~520 km. So this radius was used to

delineate between the environmental forcing and the system core forcing. There are fluctuations of the area of the upper level cloud and how dense the cloudy air is as will be seen in section 4.4, nevertheless this simple approach does seem to give a reasonable perspective on the relative importance of radiative forcing in the environment versus in the cloudy air. We should not expect the responses to the heating/cooling fields to be exactly linear however. So adding the separate responses of the environmental cooling only experiment to the system core forcing only experiment is unlikely to give exactly the same result

as the combined response.

        Figure 9 shows results of the potential temperature and vapour mixing ratio perturbations at 64 h for the environmental forcing only and system core forcing only experiments. Comparing Fig. 9a and c with Fig. 8c it appears that to a large extent the potential temperature perturbations below the canopy top in the system core for Experiment 3 are due to the environmental forcing. For the core forcing experiment there is some cooling between 5 km and 10 km in the system core

and in the environment as well, however this happened in the first 36 h of the simulation when the full forcing was imposed. This cooling in this layer can be seen several hours later at 40h for Experiment 3 in Fig, 6c. The core forcing shows stronger cooling at 13 km than at 10 km, which would seem to be because the peak longwave cooling at the canopy top occurs slightly above the peak shortwave heating, and also because the shortwave heating appears to be deeper.

        For the environmental forcing only experiment the vapour perturbation is very large at low levels and has a secondary

peak at mid-levels. The environmental forcing causes ascent in the core and while it is weaker at low-levels than upper levels (not shown) the vapour perturbations are larger because the vertical gradient of water vapour is large at low-levels. The core forcing only experiment shows a significant moistening at mid levels and this is a response to the longwave warming in the core, which produces ascent. There is a dramatic drying at low levels between radii of 160 km and 400 km, which must be due to descent of drier air from aloft. This appears to be mainly due to low-level radiative cooling in this region seen in Figs.


7a and 8a. It seems that the response to environmental forcing largely counters this drying when the total radiative forcing is applied (Fig. 8f).

## 4.3 Cloud experiments

A detailed description of the results of the cloud experiments is given in the Supplement. It is found that the changes to water vapour mixing ratio and temperature induced by nocturnal radiative forcing between 57 h and 69 h enhances the strength of a convective cell considerably, and therefore is likely to be the cause of the large increase in hydrometeor mass in this time period shown in Fig. 1b. Moreover it is found that the radiatively induced cooling aloft caused by adiabatic ascent has a bigger effect than the moistening aloft. While the vapour change causes only a moderate increase in precipitation when acting alone it is notable that in conjunction with the temperature change it can lead to a very significant increase. The experiments also demonstrate that the radiative forcing in the relatively cloud-free environment is responsible for most of the increase in convective intensity.

Cloud experiments that examined if the downturn in convective activity occurring after 48 h in Fig. 1b is due to daytime radiative forcing surprisingly did not show any suppression in the strength of a convective cell. However it is worth noting that there is a loss of low level ascent during the daytime (Fig. 7b) compared to the nighttime (Fig. 6b) so it is less likely that the daytime radiative forcing will trigger new convection at low-levels. An Additional factor, which is examined in the next section, is the effect of cold pool outflows on reducing CAPE and influencing subsequent convective activity.

## 4.4 Description of the diurnal cycles

Figure 10 shows Hovmöller diagrams of surface based CAPE, longwave up radiation at z=20.1 km and shortwave radiation down at z=20.1 km. CAPE initially increases significantly around the radius of maximum surface winds, which cause strong surface moisture and sensible heat fluxes. During the first night CAPE begins to build everywhere except near the centre and this can be mainly attributed to cooling aloft due to longwave radiation and in part to surface fluxes. At the end of the first night there is a decrease in CAPE beginning at a radius of ~100 km and this broadens with time. Nicholls and Montgomery (2013) found that typically convection would begin quite close to the centre of the large scale circulation and that spreading surface outflows would tend to trigger new convection further from the centre. This convective activity then works its way outwards, sometimes as rings, or bands of convection, with CAPE being diminished over a broad area as the warm and moist boundary layer air was replaced by cold and dry convective outflows. CAPE would gradually build as the boundary layer recovered and then the process would start again. With radiation included these oscillations showed a strong diurnal response, although there was sometimes a small oscillation of convective activity during the daytime, which was hypothesized to be due to the boundary recovering by the middle of the day. Fig. 10a shows a similar behaviour. At 43 h during the middle of the second night CAPE starts to decrease at a radius of 100 km. There is some consumption of CAPE inwards with time, as well as an outward consumption, which reaches a radius of 230 km by 55 h. CAPE then builds up at a radius of 100 km early in the third night and starts to be consumed shortly thereafter. CAPE increases significantly over a




broad area during the third night, and then is consumed as the convection works outwards, reaching a radius of over 400 km by 80 h. There is a small increase in CAPE between 66 h at a radius of 100 km and 77 h at a radius of 180 km, which may be responsible for the small peak in total hydrometeor mass at 78 h for radius less than 200 km seen in Fig. 1b.

Prior to considering the outgoing longwave radiation we show in Figure 11 horizontal sections of CAPE at 12 h intervals from 28 h to 88 h for grid 2, which reveal more details of the behaviour of CAPE. At 28 h the largest values of CAPE occur at the radius of maximum winds and convective cells begin to develop. By 40 h, which is in the middle of the second night peak values of CAPE have increased in the region of strong winds but there are now areas of lower CAPE due to convectively produced surface outflows. By 52 h, which is in the middle of the daytime, the outflows are spread out over a large circular area of radius ~210 km giving lower values of CAPE on average and convective activity in this area is close to a minimum (Fig. 1b). CAPE then builds up over a broad area (also see Fig. 10a) and by 64 h convection is starting to work out again from near the centre. By 76 h CAPE has decreased over a very large area. It is notable that there are much larger patches of decreased CAPE than occurred 24 h earlier indicating that the cold and dry outflows have increased in scale. At 88 h when the system is now a strengthening tropical storm there are spiral bands of high and low CAPE evident. Moreover, CAPE has increased significantly in the outer region of grid 2.

The longwave up shown in Fig. 10b is clearly linked with the diurnal cycle of CAPE. As CAPE is consumed during the second night producing an extensive area of upper level cloud there is a large deficit of outgoing longwave in the core between r=70 km and 100 km, and this coincides with the area of reduced CAPE. At the end of the third day there is a decrease in the longwave up as the upper level cloud dissipates. At the same time the surface fluxes cause CAPE to start to build up again. CAPE further increases during the beginning of the third night over a large area, which is caused both by surface fluxes and radiative forcing that results in cooling aloft. This cooling aloft in the system core below the canopy top is primarily a result of induced upward motion driven by differential radiative forcing. As CAPE is consumed later in the night and during the following day an extensive cloud canopy develops and a deficit in outgoing long wave occurs over a much larger area than it did previously. Moreover the radius of the deficit increases with time coinciding quite well with the increasing radius of reduced CAPE. At the end of the fourth day there is more of a deficit of outgoing radiation at large distances from the centre where deep convection has occurred more recently.

Figure 12 illustrates more detail of the outgoing radiation by showing horizontal sections from 54 h to 82 h for grid 2 (note an animation of longwave up radiation can be downloaded in the *Data availability* section). At 54 h, which is in the early afternoon of the third day, there is still quite an extensive canopy even though deep convection has decreased and there is a significant reduction of longwave around 150 km from the centre. By 60 h, which is early in the third night, the deficit of longwave radiation has reduced in the system core. By 66 h there is a significant deficit, which expands outwards with time. The deficit becomes quite patchy at 78 h and by nightfall at 82 h a patchy ring exists at a radius of approximately 400 km. Infrared imagery of mature tropical cyclones have found cyclical pulses that propagate radially outward at speeds of 5-10 m s$^{-1}$ to distances of several hundred kilometres (Dunion et al., 2014). These cold cloud top temperatures have a ring like appearance that form around sunset and dissipate during the afternoon of the following day. A modeling study by Dunion et



al. (2019) produced a similar feature and was found to be associated with radially propagating deep convection that had squall-line-like features. There appears to be significant similarities to the diurnal cycle occurring in this tropical cyclogenesis simulation, however there are some differences. In this case a ring forms later around nightfall approximately 400 km form the centre, but is very patchy and does not clearly propagate as a ring from a small radius. Rather there is a

broadening of the area of decreased outgoing longwave until the ring forms.

Horizontal sections of the hydrometeor mixing ratio at z= 11.7 km for grid 2 in Figure 13 provide more insight into the behaviour of the outgoing radiation field. At 66 h, there is a double-ring structure. It developed due to convection intensifying around the radius of maximum winds at ~60 h, followed several hours later by the formation of an outer ring of deep convection around 300 km from the centre. This outer ring developed at approximately the same location that

convection had previously reached from the first burst of convection. There was a weak low-level discontinuity of surface winds, temperature, and water vapour at this location, and moreover at this boundary there had continued to be occasional weak convection throughout the daytime hours. Consequently a double-ring structure developed for a few hours, but as the inner ring moved outwards it encountered the cold and dry pools of air from the outer ring of convection and subsequently decayed. The widespread convection due to the double-ring structure resulted in a broad decrease of outgoing longwave seen

in Figs. 12c and d. A fairly well defined outward building ring of convection is evident at 72 h, but by this time it is near the boundary of the fine grid. Convection in the second grid is not very well resolved and probably contributes to the patchiness of the convection and the longwave up radiation field at 78 h and 82 h. A significant portion of the cloudiness that occurs in a ring-like structure outside the fine grid seen in Fig. 13d is probably due to advection of hydrometeors by the strong upper level outflow.

Figure 14 focuses on the upper level outflow that develops during the second convective burst with the first two panels showing results at 67 h for Experiment 3 that has imposed radiative forcing. It can be seen that the radiative forcing acting alone produces a ring of outflow just above a height of 10 km with peak velocities of ~6 m s$^{-1}$, which has almost reached the boundary of grid 2. Note that at 62 h this ring was located at a radius of 425 km, giving a propagation speed of ~5.5 m s$^{-1}$. The following two panels shows results for Experiment 1, which also shows a ring of outflow at this level. So it appears that

this peak of outflow velocities just above 10 km at this radius is caused by a superposition of the outflow caused by radiative forcing and a relatively weak outflow caused by previous convection. At this time convection has intensified significantly from a few hours before and this is starting to drive a deeper upper-level outflow nearer the centre. Fig. 14e shows that by 68 h, at a height of 11.7 km, the outflow has reached a radius of just over 400 km. By 400 km this outflow has expanded rapidly and a double-ring structure is discernable, which is presumably a result of the double-ring of convection that occurred earlier

(Fig. 13a).

Davis and Ahijjevych (2012) found several quasi-diurnal cycles of cloud top temperature occurred preceding the formation of two tropical cyclones that they analysed. The radial extent of the cold cloud approached 600 km, which is similar to our simulations. They noted that the cold cloud signature dissipated nearly simultaneously at all radii. For this study the upper level cloud at large radii dissipates a few hours after the cloud near the centre. The real cases they studied





occurred in much more complex environments. For instance there was vertical wind shear that led to misalignment of the low-level and mid-level circulations prior to tropical cyclogenesis.

## 4.5 Summary of the physical mechanisms causing the diurnal cycles

Here we summarize the physical mechanisms that appear responsible for causing the diurnal cycles that occur in this tropical
cyclongenesis simulation. Early on in the simulation there is significant cooling aloft throughout the domain primarily due to clear-sky longwave cooling at night, which causes CAPE to increase and brings the air closer to saturation. Surface fluxes of moisture and sensible heat additionally cause CAPE to increase particularly at the radius of maximum winds and in conjunction with low level convergence caused by surface friction promote deep convective cells to form at the beginning of the second night. A canopy starts to develop at upper levels during the middle of the second night around 40 h. Once a
canopy forms the nocturnal radiative forcing still causes cooling to occur within the region now occupied by clouds, but instead of it being caused by in situ longwave cooling it is now caused by environmental longwave cooling that induces weak ascent within the cloudy air. This is mesoscale ascent leading to adiabatic cooling in a region that is punctuated by much stronger convective scale vertical motions. Additionally this weak ascent transports moisture aloft causing the relative humidity to increase, which is also favourable for convection. The convection produces cold outflows at the surface, which
play a role in initiating new convection at larger radii. Therefore the convection expands outwards with time. By the late morning CAPE has decreased significantly over a larger circular area and the values of CAPE in the relatively undisturbed air outside this area are not large enough to support deep convection. Consequently there is a lull in convective activity during the afternoon.

Following this first burst in convective activity a mid-level vortex begins to develop during the daytime. The lull in
convective activity allows more longwave radiation to escape to space in the cloud disturbance. The radiative forcing in the daytime is generally positive between middle to upper levels in the tropical disturbance, but it doesn't lead to a significant increase in stability except near the top of the canopy. Instead the radiative forcing causes ascent, which is also aided by weak radiative cooling of the environment at upper levels. This ascent in the system core results in weak adiabatic cooling as well as a small amount of moistening.

At nightfall (57 h) CAPE begins to increase significantly over a broad region, except near the centre. This increase is to a large degree because the longwave cooling of the environment, which not only results in a significant temperature decrease between the middle and upper troposphere of the environment but also of the core due to induced ascent. It appears that CAPE remains small in the centre because the system scale low-level inflow has caused cold pools to be advected towards the centre and surface fluxes at this location are relatively small due to weak surface wind speeds. Another factor is that there
is now significant warm air aloft at the centre associated with the development of the mid-level vortex that could also be leading to reduced values of CAPE. Convection intensifies around the radius of maximum winds at ~60 h where the surface fluxes of moisture and sensible heat are largest. Several hours later an outer ring of deep convection forms around 300 km from the centre as discussed in the previous section. A double-ring of convection forms resulting in a broad decrease of




outgoing longwave radiation. As the convection builds outwards it eventually reaches the edge of the fine grid (~72 h), when the ability of the model to adequately resolve convection at larger radii becomes an issue. Like the first convective burst this second also produced an expanding region of decreased CAPE, which grew to be considerably larger in size. Moreover it produced a strong outflow at upper levels. Interesting it was preceded by a maximum in peak outflow velocities just above

10 km, apparently associated with the transverse circulation caused by radiative forcing.

## 5 Conclusions

A simulation of tropical cyclogenesis was conducted with the radiation scheme included and compared to a simulation without radiation. Tropical cyclogenesis took three days for the simulation with radiation whereas the simulation without radiation did not undergo tropical cyclogenesis even after six days. Leading up to tropical cyclogenesis the simulation with

radiation showed two large bursts in convective activity separated by approximately twenty-four hours. Convective activity during the bursts peaked several hours after daybreak. Following the first burst a mid-level vortex formed during the daytime hours. Tropical cyclogenesis occurred at around the peak of the second convective burst.

In order to clarify how radiation causes the accelerated rate of development the radiative forcing fields were written out to data files at hourly intervals. These were then read in during a new simulation without radiation, microphysics and surface

fluxes activated, and imposed as a source term in the thermodynamic equation. Early on in the simulation the relatively cloud-free radiative forcing caused a horizontally homogeneous increase in CAPE and relative humidity. This was clearly a contributing factor as to why deep convection began to occur during the second night in the full physics simulation (Experiment 1), in addition to the strong sea surface fluxes of moisture and sensible heat at the radius of maximum winds. During the second night differential radiative forcing began generating deep ascent within the system core. This adiabatic

ascent caused the temperature between middle to upper levels to continue to decrease in the core at the same rate as in the environment, as well as transporting moisture aloft. During the daytime the differential radiative forcing was reduced particularly at low levels and there was no longer low-level ascent in the core, although aloft it continued to be significant. During the third night the differential radiative forcing again caused low-level ascent.

Two experiments were performed that separated the imposed radiative forcing into separate environment and core

components. These experiments demonstrate that those thermodynamic changes in the core induced by the full forcing that appear to be responsible for enhancing convection, were mainly a response to the environmental forcing.

To further understand how changes in the vertical profiles of vapour and temperature in the core brought about by radiatively induced ascent are likely to influence convection several idealized cloud experiments were conducted that were initialized with a low-level warm and moist bubble. The main focus of these experiments was the second burst of convection

that directly preceded tropical cyclogenesis. Results indicate that induced cooling in the system core had a larger impact than induced moistening, but that both acting together produce an impact that is more than the sum of their parts. Cloud experiments were also performed to examine if daytime radiative forcing significantly supressed convective activity. This





was found not to be the case. This raised the question of what was causing the decreased convective activity during the daytime.

Examination of CAPE showed it was undergoing diurnal cycles. CAPE would be consumed starting from around the radius of maximum winds and convection would then build outwards leaving a circular region of reduced CAPE where the boundary layer air had been replaced by cold and dry surface outflows. So by the following day CAPE had been reduced over a large area and this is a major factor that explains why daytime convection was much less. When CAPE is consumed it occurs at a faster rate than it is being replenished, which why there is a subsequent lull of convective activity. Fields of outgoing longwave also show diurnal cycles. Following the second burst of convection a patchy ring of reduced outgoing longwave was evident during the late evening at a radius of 400 km from the centre of the domain.

The outflow aloft caused by the radiative forcing was shown to result in a significant anticyclonic circulation. It is notable that there was an outflow driven by radiative forcing during the third night evident just above 10 km as a narrow propagating circular band. This was also evident in the full physics simulation (Experiment 1) and it preceded the deeper outflow that extended higher in the troposphere that was driven by convective scale latent heat release.

This study provides strong support for the hypothesis that nocturnal differential radiative forcing between a tropical disturbance and its relatively cloud free environment can significantly accelerate tropical cyclogenesis. Moreover it has shown how the radiative forcing creates diurnal cycles, and how these cycles are also related to the rapid consumption of CAPE by convection that builds outwards from near the centre, leaving a boundary layer that needs time to recover from modification by cold and dry outflows. It emphasizes the need to realistically parameterize cloud-radiative interactions in operational forecast models. However this simulation is very idealized and further research is required to investigate the affects of radiation on tropical cyclogenesis in more complex environments, particularly those having vertical wind shear or those occurring in easterly waves (e.g. Dunkerton et al., 2009; Wang, 2012).

*Data availability.*

A longwave up animation is located at: http://cires1.colorado.edu/nicholls/data3/LWUP_animation/LW_Animation.avi

*Author contributions.* MN, WS and RP designed the experiments. SS and MN updated the model code. NW checked results of the radiation scheme by comparing with another radiation algorithm. MN conducted the model simulations. MN and WS analysed model data. MN prepared the manuscript with contributions from co-authors.

*Acknowledgements.* We are extremely grateful to Saurabh Barve for computational assistance. This research was supported by the National Science Foundation, under Grant NSF AGS 1445875.



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



**Table 1**. List of Experiments

| Experiment | Description |
|---|---|
| 1 | Full physics simulation with radiation |
| 2 | Full physics simulation without radiation |
| 3 | Imposed radiation |
| 4 | Imposed radiation with the environmental forcing only |
| 5 | Imposed radiation with the system core forcing only |
| 6 | Cloud simulation with the averaged vertical profile at 57 h for the full physics simulation |
| 7 | Cloud simulation with nighttime radiative forcing modification to both vapour and temperature (57 h - 69 h) |
| 8 | Cloud simulation with nighttime radiative forcing modification to vapour only (57 h - 69 h) |
| 9 | Cloud simulation with nighttime radiative forcing modification to temperature only (57 h - 69 h) |
| 10 | Cloud simulation with nighttime radiative forcing modification with the environmental forcing only (57 h - 69 h) |
| 11 | Cloud simulation with nighttime radiative forcing modification with the core forcing only (57 h - 69 h) |
| 12 | Cloud simulation with the averaged vertical profile at 47 h for the full physics simulation |
| 13 | Cloud simulation with daytime radiative forcing modification to both vapour and temperature (47 h - 55 h) |

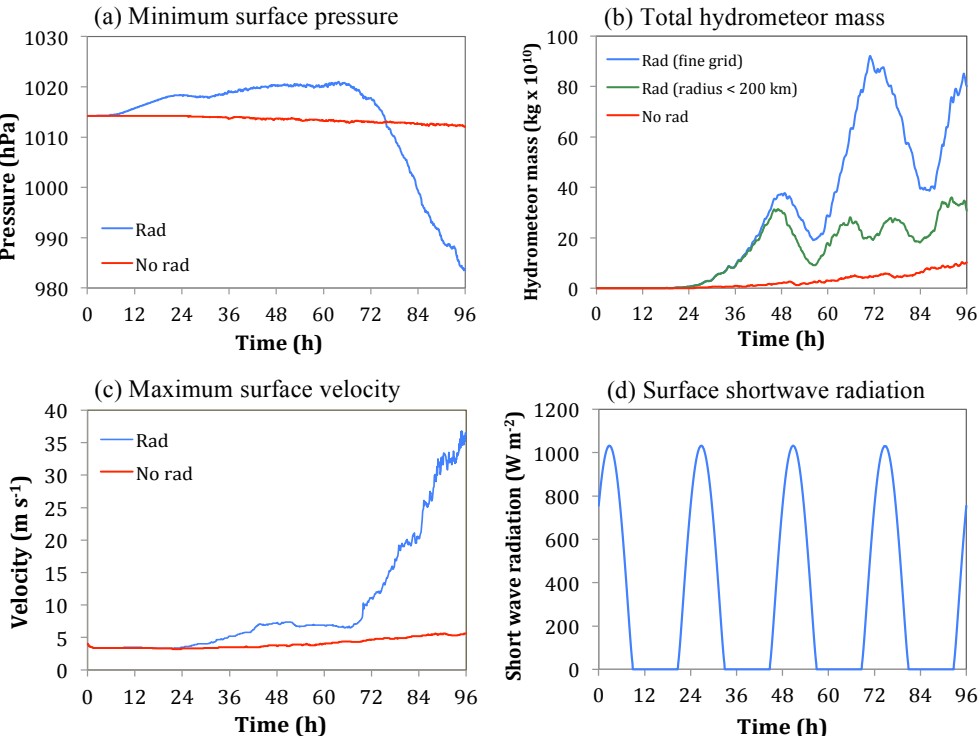

**Figure 1**. Time series for the full physics simulation with radiation (Experiment 1) including comparison with the no radiation case (Experiment 2). (**a**) Minimum surface pressure, (**b**) total liquid and ice mass content in the fine grid domain, and in a radius less than 200 km for Experiment 1, (**c**) maximum azimuthally averaged surface tangential velocity and (**d**) clear-sky shortwave radiation at the surface.

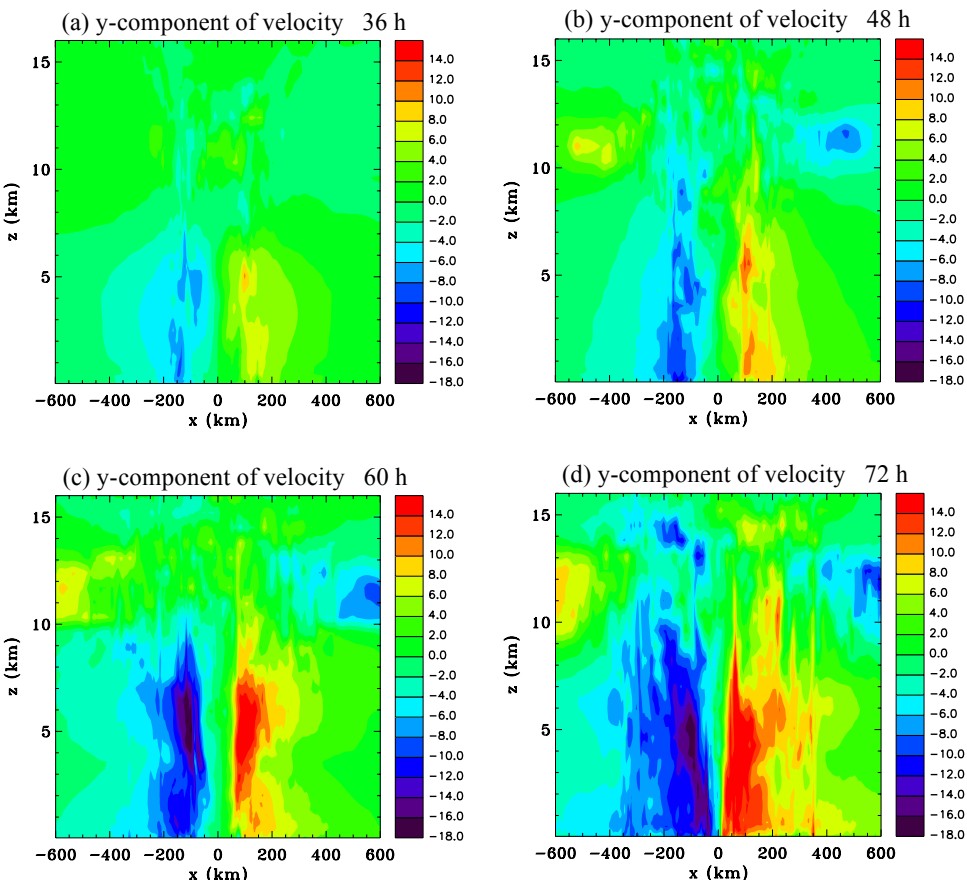

**Figure 2**. Velocity in the y-direction through the center of grid 2 for Experiment 1 (m s⁻¹). (**a**) 36 h, (**b**) 48 h, (**c**) 60 h and (**d**) 72 h.



**Figure 3**. Results for Experiment 1 at 45 h for grid 2. **(a)** Radiative flux convergence ($\times 10^{-5}$ K s$^{-1}$), **(b)** radiative flux convergence at z=6.5 km ($\times 10^{-5}$ K s$^{-1}$), **(c)** x-component of velocity (m s$^{-1}$), **(d)** hydrometeor mixing ratio at z=11.7 km (g kg$^{-1}$), **(e)** potential temperature perturbation (K), and **(f)** relative humidity. Note vertical sections are through the center of the domain.

**Figure 4**. Results for Experiment 1 at 52 h for grid 2. (**a**) Radiative flux convergence ($\times 10^{-5}$ K s$^{-1}$), (**b**) radiative flux convergence at z=6.5 km ($\times 10^{-5}$ K s$^{-1}$), (**c**) x-component of velocity (m s$^{-1}$), (**d**) hydrometeor mixing ratio at z=11.7 km (g kg$^{-1}$), (**e**) downward shortwave radiation (W m$^{-2}$), and (**f**) upward shortwave radiation (W m$^{-2}$). Note vertical sections are through the center of the domain.



**Figure 5**. Results for Experiment 1 at 69 h for grid 2. (**a**) Radiative flux convergence ($\times 10^{-5}$ K s$^{-1}$), (**b**) radiative flux convergence at z=6.5 km ($\times 10^{-5}$ K s$^{-1}$), (**c**) x-component of velocity (m s$^{-1}$). (**d**) hydrometeor mixing ratio at z=11.7 km (g kg$^{-1}$), (**e**) potential temperature perturbation (K), (**f**) relative humidity, **(CONTINUED NEXT PAGE)**



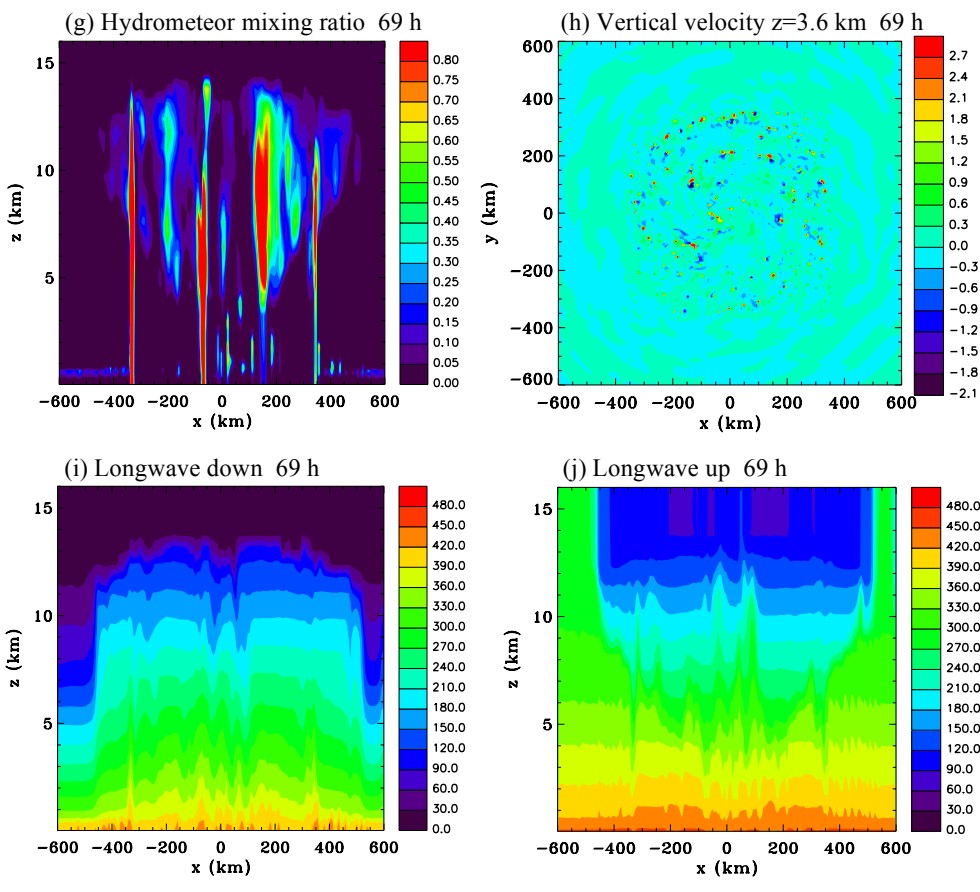

**Figure 5 continued:** (**g**) hydrometeor mixing ratio (g kg$^{-1}$), (**h**) vertical velocity at z=3.6 km (m s$^{-1}$), (**i**) downward long wave radiation (W m$^{-2}$), and (**j**) upward long wave radiation (W m$^{-2}$). Note vertical sections are through the center of the domain.

**Figure 6.** Azimuthally averaged vertical sections at 40 h for Experiment 3. The centre of the domain is at r=0 km. (**a**) Radiative flux convergence ($\times\ 10^{-5}\,\mathrm{K\ s^{-1}}$), (**b**) vertical velocity ($\mathrm{m\ s^{-1}}$), (**c**) radial velocity ($\mathrm{m\ s^{-1}}$), (**d**) tangential velocity ($\mathrm{m\ s^{-1}}$), (**e**) Potential temperature perturbation (K), (**f**) vapour mixing ratio ($\mathrm{g\ kg^{-1}}$), (**CONTINUED NEXT PAGE**)



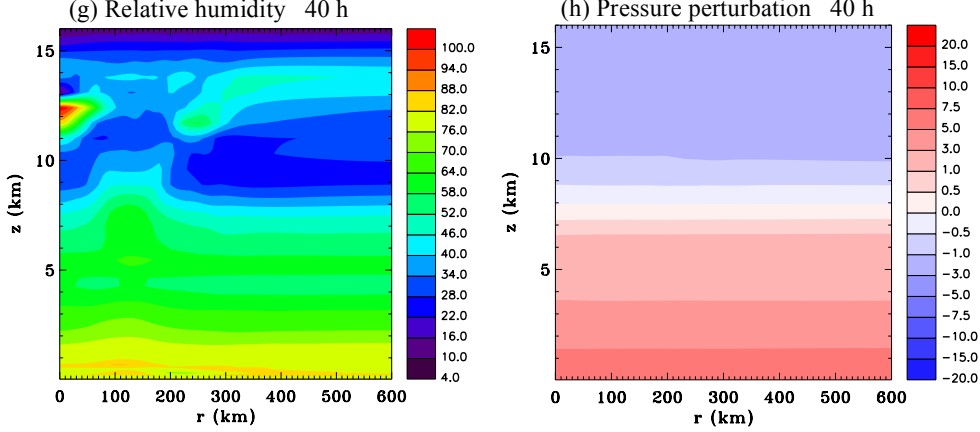

**Figure 6 continued:** (**g**) relative humidity and (**h**) pressure perturbation (hPa).



**Figure 7.** Azimuthally averaged vertical sections at 52 h for Experiment 3. The centre of the domain is at r=0 km. (**a**) Radiative flux convergence ($\times 10^{-5}$ K s$^{-1}$), (**b**) vertical velocity (m s$^{-1}$), (**c**) radial velocity (m s$^{-1}$), (**d**) tangential velocity (m s$^{-1}$), (**e**) Potential temperature perturbation (K), (**f**) vapour mixing ratio (g kg$^{-1}$), **(CONTINUED NEXT PAGE)**



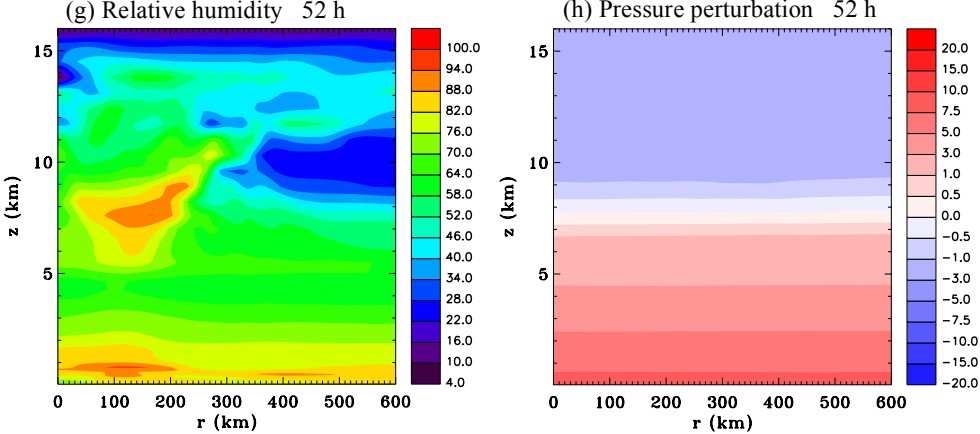

**Figure 7 continued:** (**g**) relative humidity and (**h**) pressure perturbation (hPa).

**Figure 8.** Azimuthally averaged vertical sections at 64 h for Experiment 3. The centre of the domain is at r=0 km. (a) Radiative flux convergence ($\times 10^{-5}$ K s$^{-1}$), (b) vertical velocity (m s$^{-1}$), (c) radial velocity (m s$^{-1}$), (d) tangential velocity (m s$^{-1}$), (e) Potential temperature perturbation (K), (f) vapour mixing ratio (g kg$^{-1}$), **(CONTINUED NEXT PAGE)**



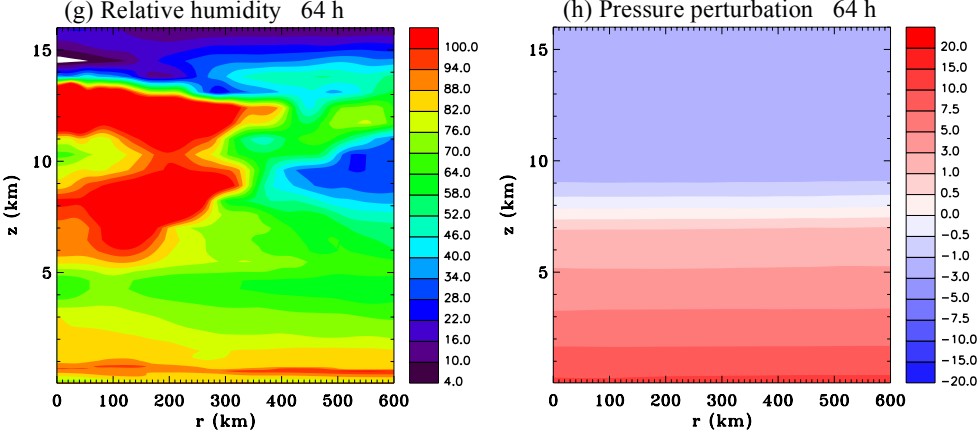

**Figure 8 continued:** (**g**) relative humidity and (**h**) pressure perturbation (hPa).

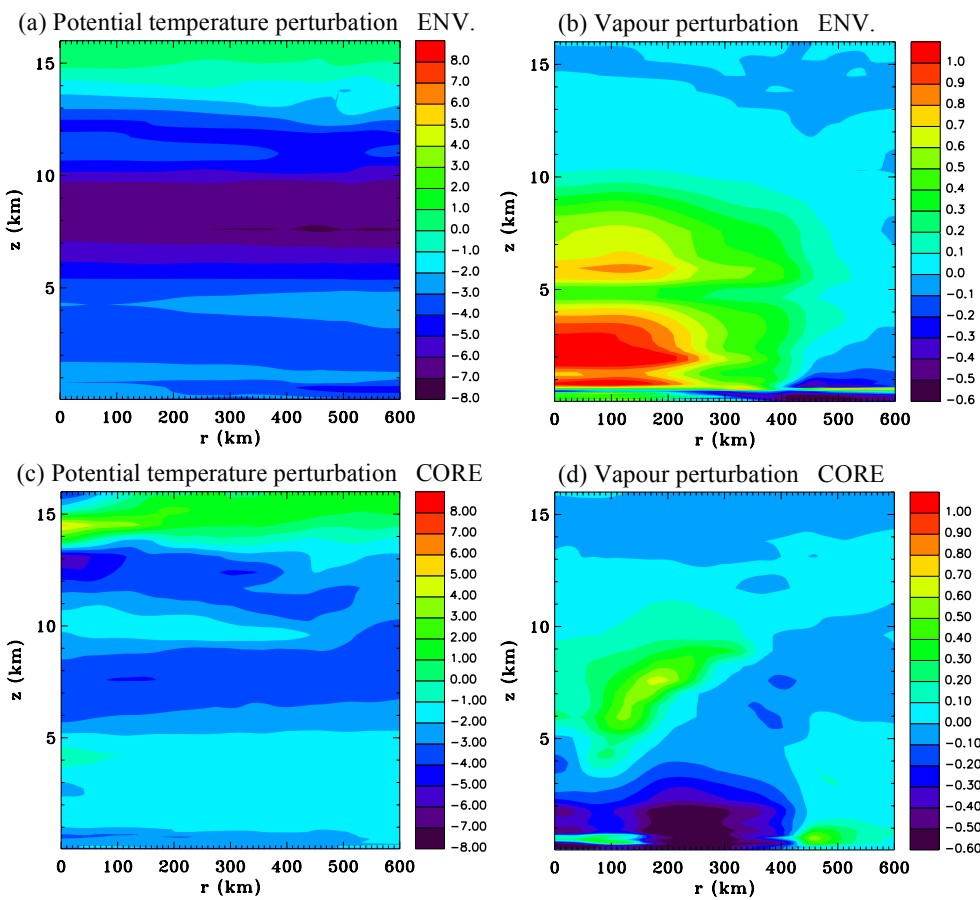

**Figure 9.** Azimuthally averaged vertical sections at 64 h for Experiment 4 and 5.
(**a**) Potential temperature perturbation (K) for Experiment 4, (**b**) vapour mixing
ratio (g kg[-1]) for Experiment 4, (**c**) potential temperature perturbation (K) for
Experiment 5 and (**d**) vapour mixing ratio (g kg[-1]) for Experiment 5.





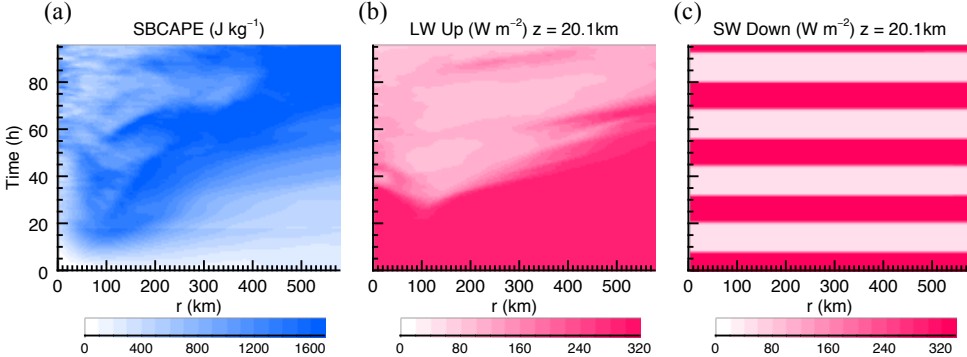

**Figure 10**. Hovmöller diagrams for Experiment 1. **(a)** Surface based CAPE (J kg⁻¹), **(b)** longwave up radiation at z=20.1 km (W m⁻²) and **(c)** shortwave radiation down at z=20.1 km (W m⁻²).



**Figure 11.** Horizontal sections of surface based CAPE (J kg$^{-1}$) for Experiment 1 at
(**a**) 28 h, (**b**) 40 h, (**c**) 52 h, (**d**) 64 h, (**e**) 76 h and (**f**) 88 h.



**Figure 12.** Longwave up radiation (W m$^{-2}$) at z=20.1 km for Experiment 1 at (**a**) 54 h, (**b**) 60 h, (**c**) 66 h, (**d**) 72 h, (**e**) 78 h and (**f**) 82 h.



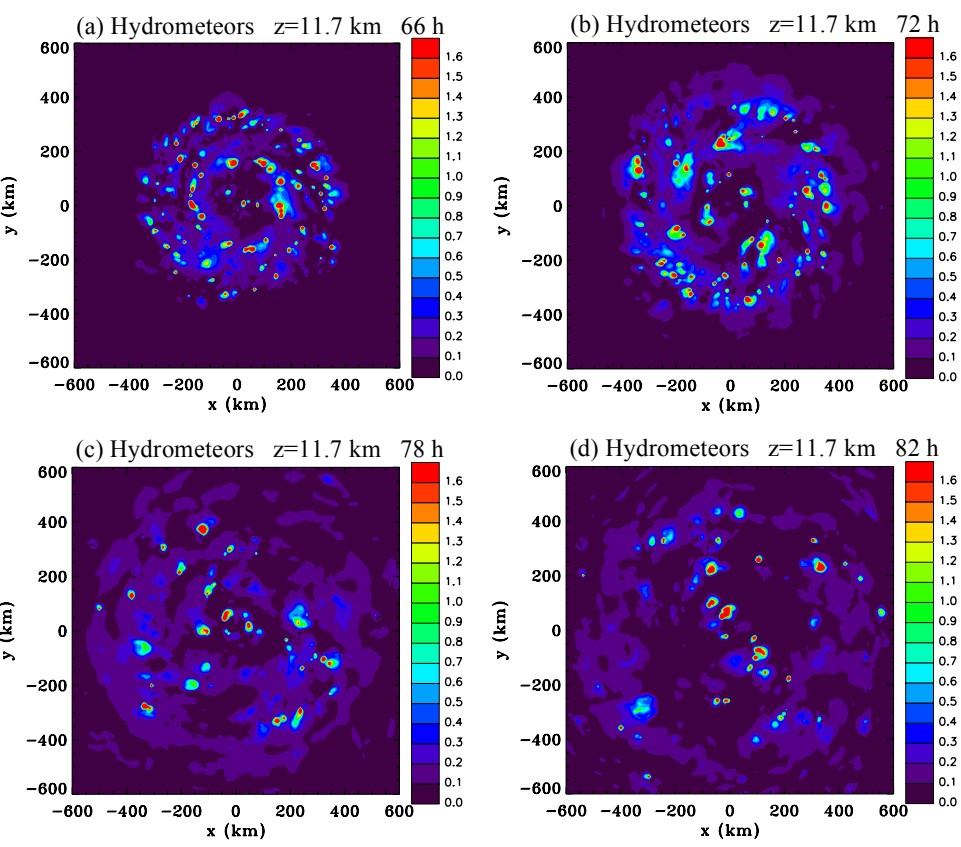

**Figure 13.** Total hydrometeor mixing ratio (g kg$^{-1}$) at z=11.7 km for Experiment 1 at
(**a**) 66 h, (**b**) 72 h, (**c**) 78 h and (**d**) 82 h.

(a) x-component of velocity  67 h  EXP. 3    (b) Radial velocity  z=10.3 km  67 h  EXP. 3

(c) x-component of velocity  67 h  EXP. 1    (d) Radial velocity  z=10.3 km  67 h  EXP. 1

(e) Radial velocity  z=11.7 km  68 h  EXP. 1    (f) Radial velocity  z=11.7 km  72 h  EXP. 1

**Figure 14.** Upper level outflows (m s$^{-1}$). (**a**) x-component of velocity through the centre of the domain at 67 h for Experiment 3, (**b**) radial velocity for z= 10.3 km, at 67 h for Experiment 3, (**c**) x-component of velocity through the centre of the domain at 67 h for Experiment 1, (**d**) radial velocity for z= 10.3 km, at 67 h for Experiment 1, (**e**) radial velocity for z= 11.7 km, at 68 h for Experiment 1 and (**f**) radial velocity for z= 11.7 km, at 72 h for Experiment 1.