# Peer review of "A numerical modelling study of the physical mechanisms causing radiation to accelerate tropical cyclogenesis and cause diurnal cycles"

_Atmospheric Chemistry and Physics, 2019_

## Referee Comment (RC1) · Anonymous Referee #1 · 19 Nov 2019

**Review of "Nicholls et al."**

The modeling study by Nicholls et al. explores the physical mechanisms behind the diurnal cycle of a developing tropical cyclone. This is explored through several sensitivity experiments that examine how diurnal cycles in radiative forcing lead to diurnal cycles in various aspects of an evolving tropical system. While such mechanisms were previously already known, they have not yet been analysed in detail.  In particular, the study shows how night radiative cooling and the resulting temperature and moisture perturbation accelerates the development of tropical cyclones.

The study has potential to broaden the knowledge of the drivers of tropical cyclone (TC) diurnal cycles. However, in my opinion the study in present form is not able to draw strong conclusions due to several deficiencies, listed below. Moreover, I am not sure what are the study's main findings considering the significant amount of previously published literature on the topic. The descriptions are often too long, and the reader gets lost in details. A shorter, sharper, conclusion section that puts the results in the context of other work on the topic may help in putting the message across in a more effective way.

**Major points**

1.) The study setup does not allow a clear separation between features of the TC diurnal cycle and those related to the intensification and the development of the TC.  I would therefore suggest separating the two problems by running one of the experiments to a statistical steady state. One could average over the course of several days and get a robust picture of diurnal changes in several TC-related quantities and properties.

   The role of diurnal cycle in the acceleration of the tropical cyclogenesis could be analysed only after understanding its role in the TCs steady state.

2.) The study does not address the possible stochastic nature of tropical cyclogenesis and its acceleration as brought up by a recent study by Ruppert and O'Neill 2019. They conclude that the diurnal cycle does not impact the timing of cyclogenesis. I would like to see a few more ensemble runs of several experiments that would determine, which of the numerous features described are robust, and which aren't (if any).

   Your study seems to imply that the diurnal cycle is the most important driving factor of the acceleration of tropical cyclogenesis. However, I am not yet convinced. I don't think you can conclude that without at least one additional experiment using daily averaged insolation.

   Would the TC develop in perpetual nighttime conditions (no insolation)?

   Would the TC develop in conditions representative of diurnal average insolation? (I assume the insolation value would be close to 300 W/m$^2$)

Would the TC develop in perpetual middle-of-the-day conditions? (insolation > 800 W/m$^2$)

3.) It is not clear, which of the findings of the study are novel and which just confirm previous work.  Can you better point out what is novel? A discussion that would compare the main results of the manuscript with previously published literature might help in addressing this point. At the same time it would be great if you could shorten your conclusions section.

4.) A large deficiency of the study is its poor graphical representation and labelling.  In particular:
   a. it is really hard to understand what local time "40 h", "45 h" etc. correspond to. especially because the simulations are started at a random time in the morning, not at midnight. The descriptions in the text are not helpful enough. I suggest that the authors add the specific local time label everywhere, where they currently have a time label, on all "snapshot-type" figures.

   b. I suggest more of time evolution plots (like Fig. 1,10) and less of snapshot-like filled contour plots. However, such time evolution figures have to clearly (in every panel) give some indications about local time or insolation (maybe limits of day/night with a shading). It is really hard to understand where exactly the day-night boundaries are in Figures 1 and 10!

5.) Several sections of the manuscript are lengthy: the reader gets easily lost in the long descriptions of the TC evolution, losing the overview of the main scientific results of the manuscript. You may want to consider decreasing the number of figures by showing only those strictly needed to support your main results.

**Additional general points:**

1.) When talking about diurnal cycles you never mention maximum wind speed, which is of a direct societal relevance. Do the surface (azimuthal) winds also respond to diurnal cycles of radiation, as shown for instance in the modelling work by Navarro at al., 2017?

2.) You often refer to the work by Gray and Jacobson, 1977. They do, however, refer to diurnal cycle of deep convection, larger deep convective systems, but not TCs.  Is there a difference between the diurnal cycle in deep convection and in TCs?  Or is the key to understanding TC cycles simply in studying deep convection?

3.) You may disregard the following comment, as it is hard to argue objectively on the use of colormaps. But, nevertheless, please "Scrap rainbow color scales!" (see Hawkins et al., 2015, Nature - https://www.nature.com/articles/519291d - Sec2)

There is a lot of evidence that jet or rainbow colormap you constantly use is not the best, so I would recommend you to consider a different colormap in your future work. More information: https://www.climate-lab-book.ac.uk/2014/end-of-the-rainbow/, https://www.wsj.com/articles/weather-forecasts-should-get-over-the-rainbow-1538054430, https://blogs.mathworks.com/headlines/2018/10/10/a-dangerous-rainbow-why-colormaps-matter/.

4.) Hurricane formation Pathway 1 vs. Pathway 2: Which of them is more representative of what observed in nature?
Is the radiation+diurnal cycle aiding cyclogenesis in the same way in both types of development?

5.) I would consider including more information on the "cloud experiments" in the main text, together with the figure S3. And at the same time prevent the manuscript from growing by shortening rest of the results, e.g. sections 4.2 and 4.4.

**Specific comments**
Note that some of the specific comments were already mentioned above.

 **Intro**

page 2, lines 11- 18:
Describe how a diurnal cycle in tropical cyclones looks like? What is changing throughout the day? How? At what local time we observe minima and maxima of the quantities like high clouds, precipitation, ice water path, liquid water path, CAPE, maximum winds, mesoscale ascent, etc.

page 2, lines 11-17:
What exactly is disputed and unknown in the cause of diurnal cycles in tropical cyclones? Could you be more specific!

page2 line 25 – page 3 line 28:
You dedicate one page to results of N15 study. Please be more concise. You never comment how the N15 results relate to the large amount of literature on the topic of diurnal cycles of TCs.

page 3, lines 11-12:
Could you better describe the simulation with no microphysics, surface fluxes, and radiation.
Does it involve cloud formation? Does it involve latent heating/cooling? Is it simply a simulation with clear air + radiative forcing from the reference (full physics) simulation?

page 3, line 18 – 28:

Does the formation pathway influence your results? Would the several mechanisms be same important?
Compared to N15, the TC development is slower, for example? Could this change the relative importance of several mechanisms?

**2 Numerical model and initial conditions**

page 4, lines 14-18:
How does the nested grid setup look like? A sketch would be useful for its easier visualization.

How is the simulation set up. More specifically:
 What are the boundary conditions of the domain?
Is the simulation following/centred on the developing hurricane?
Or how do you make sure the perturbation is in/near the middle of the domain?
Are large-scale winds prescribed? Is wind shear present?

**3 Description of experiments**

Experiment 3: I assume latent heating/cooling is also not allowed to occur (due to no clouds, no microphysics)? Please describe that experiment in some more detail.

**4 Results**

**4.1 General description of Experiment 1 and comparison with Experiment 2**

Is a 6-day-long simulation enough to support some of the conclusions? Would a TC still develop in the NoRad experiment if the simulation would run for longer?

page 6, lines 12-14
What is causing that double peak? At what local time are those peaks centred? Is there a role for SW heating of clouds promoting anvil cloud maintenance? (e.g. Ruppert and O'Neill 2019, Ruppert and Klocke, 2019)

Is Figure 2 needed? I don't see much more information from the several contour plots compared to the time evolution plots in Fig. 1.

Fig. 3 – a,c,d,e: What's the advantage of a noisy cross section compared to a (probably) more smoothed and therefore easier-to-understand radial plot (i.e. azimuthally averaged)?

page 6, line 28:
"hole in the cloud canopy": is this just a stochastic/random feature? Is it important or not? If not, I would remove it from the text.

You are not comparing Exp1 with Exp2. Either remove that from the title of the section 4.1 or add the description! (maybe not needed though, as the differences

between a full simulation setup and "No Rad" seem to be already known based on N15 and other work).

page 6, line 33:
What local time is that?
Please, when relevant, add a local time to the descriptions or plot titles!

page 7, line 5:
"The shortwave down" => The downwelling shortwave radiation
Please correct other instances like that too. Same for "longwave down/up"

Remove panel 4f as it does not give any additional information compared to 4e.

page 7, line 12:
What caused the cooling at 8 km: sublimation?

I don't think you need to show the vertical velocity cross section at 3.6 km (fig5h). The hydrometeor cross sections already confirm the existence of deep convective storms.
What is the purpose of the panel 5g?
Also, please remove one of the panels (i) and (j), as they basically show the same information.

page 7, lines 13-16:
I think you could either remove or better explain (if you find it important) the section starting with "The horizontal cross-section…..above the cloud canopy".

**4.2 Imposed radiative forcing experiments**

Why are the changes horizontally homogeneous if they were not homogeneous in Exp 1? I thought you simply save the radiative fluxes from Exp 1 and read those in to drive Exp 3?
Why does Exp 3 not appear in Fig. 1? Is its time evolution very similar to Exp 1? If not, what causes the differences?

line 21: "relatively cloud-free"
I thought your Exp 3 has no clouds? Does no microphysics mean no clouds? How do you condense water if microphysics is turned off? Please explain your setup better.
If there are clouds, than how do we evaluate the contributions of latent heating/cooling vs radiative heating/cooling?

Could you add a time evolution plot of CAPE. I am interested how the time evolution of CAPE in Exp 3 compares with Exp 1. Could you put the CAPE numbers in context. What does a doubling of CAPE means? Isn't a CAPE of 400 still considered as relatively small?

page 7, line 24:
Why at hour 40 and not at hour 45 as in the previous example. If there is a

reason for choosing middle of the night rather than morning conditions, we need to know about it.
Therefore we cannot directly compare the panels 3e,f with 6e,g.

Moreover, could you switch panels f and g in Fig. 6 for consistency.

page 7, line 31:
What about latent heating? What is the role of latent heating? You only talk about changes brought by radiative forcing, but does this explain the full evolution? Is radiative forcing fully driving all of the storm evolution, including latent processes?

*"These mesoscale changes to the vertical profiles of vapour and moisture brought about by radiative forcing are likely to be the main reason for the increase of deep convection during the nighttime seen in Fig. 1b. "*

Could you prove that? Could you have a simulation with/without the additional vapour perturbation? (or something similar)

page 8, lines 2-4: *The pressure perturbation...*
Do you really need a separate panel in the figure for that? I would suggest removing it, as we already know about it from the description of the time evolution of pressure in Fig. 1. On that note, it would be interesting to note if the evolution in Exp 1 and 3 are the same or if they differ.

Also, are we supposed to compare Figures 4 and 7? If so, could you also show a radial representation (axisymmetric average) of Fig. 4?

Could you summarize what the main finding is from Exp 3. I am still not totally sure about it. By prescribing the radiative fluxes we can get the evolution of a TC, together with its diurnal cycles?

Page 8, 9:
I started to lose track after about line 15 on page 8 till line 8 on page 9. It does seem to be a bit repetitive. Why do we need to have the discussion on the night processes twice (related to figure 8 and 6)?.

Could you think of ways of condensing the information from several simulations in one figure, which shows the time evolution? It is not easy to follow the details of the time evolution by switching between several contour plots.

Page 9, lines 10 -12:

*"An interesting question is how much the changes are a result of radiative forcing in the relatively cloud-free environment versus in the cloud disturbance? To address this question Experiments 4 and 5 were conducted "*

But I thought you addressed this issue already in N15?  I thought this is the central result of N15 and therefore does not need to be revisited here?

**S1 Cloud experiments (and related section 4.3)**

I don't understand at what conditions experiments 6-13 are run?
Are 6-11 performed in the absence of shortwave radiation?
Are 12,13 performed in presence of insolation?
How strong, what is the insolation value?
What are the instantaneous radiative fluxes doing?
How important would be the instantaneous radiative forcing compared with changing the initial conditions as you do in Exp 7-11,13?
Would an Exp6-like simulation performed at no insolation (night-time conditions) vs. another Exp6-like simulation performed now at 1000 W/m$^2$ insolation be significantly different? Why/why not?
Moreover, I assume that the top of the atmosphere radiative fluxes, in particular the outgoing longwave radiation, could have a different behaviour compared to the accumulated precipitation.

What is the domain size? Is this a smaller domain, with only a single convective plume, arising from the moist bubble, as the Fig. S2 suggests?

Finally, do your result conclude that the diurnal cycle of tropical convection and tropical cyclones behave the same way? Are there any significant differences between a single convective plume and a large organized tropical cyclone?

Exp 6 vs. 9: What causes CAPE to increase?

**4.4 Description of diurnal cycles**

page 10, line 18:
longwave up radiation = outgoing longwave radiation
shortwave radiation down = insolation
(I understand there are differences between the top-of-the atmosphere and 20 km fluxes, but I guess those are negligible for the purpose of you study)

I am in general grateful for any time evolution plot, particularly such in which the x axis is the radial dimension away from the storm center.
However, panels a and b in Figure 10 are not helping much in supporting the discussion in the beginning of section 4.4. I CANNOT see much of the CAPE fluctuations described.

You should also probably remove the panel (c) and rather draw some sort of limits of day/insolation on panels (a) and (b). That would help the reader much more than a separate panel showing insolation. Moreover, please significantly expand the y axis for better visualization.

Due to reasons described above I avoid commenting on the behaviour/oscillations (?) of CAPE.

page 11, line 15:

*"The longwave up shown in Fig. 10b is clearly linked with the diurnal cycle of CAPE."*
I cannot see that from Fig 10a,b. So either the link is not that straightforward, or the visualization needs to be improved to support your statements.

An alternative way to a good visualization would be to calculate some for of correlation between the two quantities (or the correlation of the anomalies from the domain mean at a certain time?).

page 11, from line 26 on:
The description of Figure 12 is extremely long, and does not seem to add too much to the main message of the paper.
Same for Figures 13 and 14.

Please shorten significantly the section 4.4, adding just the information needed to support the key findings of the manuscript.

In addition, the last paragraph of section 4.4 seems to be out of place.

**4.5 Summary of physical mechanisms**

page 13, line 6: "*causes CAPE to increase*"
Would it be correct to say (if I understand it correctly): clear-sky cooling destabilizes the atmosphere causing CAPE to increase, and…

page 13, line 9: "*a canopy*" - could you be more specific? Why don't you simply say – clouds/high clouds/(high) cloud canopy?

page 13, line 10-12:
*"Once a canopy forms the nocturnal radiative forcing still causes cooling to occur within the region now occupied by clouds, but instead of it being caused by in situ longwave cooling it is now caused by environmental longwave cooling that induces weak ascent within the cloudy air. "*

This all makes physically sense to me, but your colormap on Figs. 3e and 5e does not allow me to verify your statement in an easy and quick way. The colormap contours between +1.5 and -2.2 are don't have enough contrast, particularly not if one prints the Figure.
Also, I would be much more confident in your statements if the pattern would be less noisy. Why don't you average it azimuthally?

page 13-14:
It would be great if you managed to come up with a summary plot, which should clearly show most of what described in section 4.5. This could be a time evolution plot.

I am worried that you are over interpreting the results, when describing some details of CAPE behavior, particularly after line 25. I am not convinced that the

behavior would be consistent between several ensemble members, if you were to analyze a few more simulations started with perturbed initial conditions.

**5 Conclusions**

*"A simulation of tropical cyclogenesis was conducted with the radiation scheme included and compared to a simulation without radiation. "*
I don't think that's the focus of your manuscript, so I wouldn't start the conclusion with that sentence.

page 14, lines 25-26: please rewrite the sentence

page 14:
Since you mention a lot the cloud experiments, you may want to think about including them back in the manuscript, and at the same time cutting short on some other descriptions and figures.

page 15, lines 10-13:
Is the statement starting with *"It is notable…"* important enough to be in conclusions?

**References:**

- Gray and Jacobson, 1977: Diurnal variation of deep cumulus convection, MWR.
- Hawkins et al., 2015: Scrap rainbow color scales, Nature
- Navarro et al., 2017: Balanced Response of an Axisymmetric Tropical Cyclone to Periodic Diurnal Heating, JAS
- Ruppert and O'Neill, 2019: Diurnal Cloud and Circulation Changes in Simulated Tropical Cyclones, GRL
- Ruppert and Klocke, 2019: The Two Diurnal Modes of Tropical Upward Motion, GRL

---

## Referee Comment (RC2) · Anonymous Referee #2 · 24 Nov 2019

**Review of "Nicholls et al."**

The authors explored the physical mechanisms for radiation to accelerate tropical cyclogenesis and impact of diurnal cycles. The manuscript's premise is interesting and important. In addition, the manuscript is generally well written, and it is relatively easy to follow the authors' logic.

Before accepting this manuscript for publication in ACP, I suggest that the manuscript would benefit from:

(1) a complete bibliography and discussion of previous work related to the numerical modeling and influence of radiation of TCs (some of recommended works are listed at the end), regardless of whether these works specifically address cyclogenesis, they address many key ideas (e.g., radiation induced destabilization, the diurnal cycle of TCs, moistening of tropical disturbances, and upper level outflow in TCs) in this paper;

(2) clarifying the innovation of the paper;

(3) improving the quality of figures and representation

I reserve comments on minor issues at this time, because the revised version addressing the major issues above should result in a substantially different manuscript.

Fovell, K. L. Bu, Y. P. Corbosiero, W. Tung, Y. Cao, H. Kuo, L. Hsu, and H. Su, 2016: Influence of cloud microphysics and radiation on tropical cyclone structure and motion. Multiscale Convection-Coupled Systems in the Tropics: A Tribute to Dr. Michio Yanai, Meteor. Monogr., No. 56, Amer. Meteor. Soc., https://doi.org/10.1175/AMSMONOGRAPHS-D-15-0006.1.

Caroline J. Muller, David M. Romps, 2018: Acceleration of tropical cyclogenesis by self-aggregation feedbacks, Proceedings of the National Academy of Sciences, 115 (12),2930-2935; DOI: 10.1073/pnas.1719967115

Navarro, E.L. and G.J. Hakim, 2016: Idealized numerical modeling of the diurnal cycle of tropical cyclones. J. Atmos. Sci., **73,** DOI: 10.1175/JAS-D-15-0349.1 .

Navarro, Hakim, and H. E. Willoughby, 2017: Balanced response of an axisymmetric

tropical cyclone to periodic diurnal heating. J. Atmos. Sci.,74 (10), 3325–3337

Tang X, Z. Tan, J. Fang, Q. Sun, and F. Zhang, 2017: Impacts of diurnal radiation cycle on secondary eyewall formation. J. Atmos. Sci., 74, 3079–3098, https://doi.org/10.1175/JAS-D-17-0020.1 .

Tang, X., Z.-M. Tan, J. Fang, E. B. Munsell, and F. Zhang, 2019: Impact of the diurnal radiation contrast on the contraction of radius of maximum wind during intensification of hurricane Edouard (2014). Journal of the Atmospheric Sciences, 76 (2), 421–432.

Tang, X., Cai, Q., Fang, J., & Tan, Z. M. (2019). Land–sea contrast in the diurnal variation of precipitation from landfalling tropical cyclones. Journal of Geophysical Research: Atmospheres. Doi: 10.1029/2019JD031454, https://agupubs.onlinelibrary.wiley.com/doi/10.1029/2019JD031454

Trabing, B. C., M. M. Bell, and B. R. Brown, 2019: Impacts of Radiation and Upper Tropospheric Temperatures on Tropical Cyclone Structure and Intensity. J. Atmos. Sci., 76, 135-153.